# Goat Milk Allergy and a Potential Role for Goat Milk in Cow’s Milk Allergy

**DOI:** 10.3390/nu16152402

**Published:** 2024-07-24

**Authors:** Olga Benjamin-van Aalst, Christophe Dupont, Lucie van der Zee, Johan Garssen, Karen Knipping

**Affiliations:** 1Noordwest Hospital Group, 1815 JD Alkmaar, The Netherlands; 2Onze Lieve Vrouwe Gasthuis (OLVG) Hospital, 1091 AC Amsterdam, The Netherlands; 3Ramsay Group, Pediatric Gastroenterology Department, Marcel Sembat Clinic, 75004 Paris, France; 4Ausnutria B.V., 8025 BM Zwolle, The Netherlands; 5Department of Pharmaceutical Sciences, Utrecht University, 3584 CG Utrecht, The Netherlands

**Keywords:** cow milk allergy, goat milk, infant formula

## Abstract

In many parts of the world, goat milk has been part of the human diet for millennia. Allergy to goat’s milk, not associated with allergy to cow’s milk, is a rare disorder, although some cases have been described. Goat milk proteins have substantial homology with cow’s milk proteins and even show cross-reactivity; therefore, they are not advised as an alternative to cow’s milk for infants with IgE-mediated cow’s milk allergies. However, there are indications that, due to the composition of the goat milk proteins, goat milk proteins show lower allergenicity than cow’s milk due to a lower αS1-casein content. For this reason, goat milk might be a better choice over cow’s milk as a first source of protein when breastfeeding is not possible or after the breastfeeding period. Additionally, some studies show that goat milk could play a role in specific types of non-IgE-mediated cow milk allergy or even in the prevention of sensitization to cow’s milk proteins. This review discusses a possible role of goat milk in non-IgE mediated allergy and the prevention or oral tolerance induction of milk allergy.

## 1. Introduction

In recent decades, the prevalence of food allergies has increased significantly (6–8% of children worldwide), including cow’s milk allergy (CMA), which is one of the most common allergies developing in childhood, with a prevalence of 1.4–3.8% of young children [1]. Although there are differences in the prevalence of food allergies such as peanut and wheat in different parts of the world, the prevalence of CMA is comparable worldwide [2]. Generally, CMA spontaneously resolves at preschool ages; however, a small number of children do not outgrow their allergy to cow’s milk (CM), and it will persist in adolescence and even adulthood [3]. CMA is an adverse reaction to proteins in CM and is mediated by an antigen-/allergen-specific immune reaction. The immune response can be immunoglobulin E (IgE)-mediated, non-IgE-mediated (cell-mediated), or mixed (IgE-mediated and non-IgE-mediated).

IgE-mediated CMA is a type I (or immediate) hypersensitivity immune response to one or more milk proteins [4]. It presents immediate classical symptoms, affecting the skin (hives and angioedema (rash on lips, eyes, or face)), gastrointestinal symptoms (vomiting), breathing difficulties (coughing and wheezing), hypotension, and sometimes anaphylaxis. A type I response is mediated by IgE antibodies that are produced by the immune system towards a specific allergen/epitope. These IgE antibodies bind to the high-affinity IgE receptor FcεRI on mast cells and basophils. Upon secondary contact with the same allergen, the allergen will bind several of the cell-bound IgE molecules, thereby cross-linking IgE molecules and, as a consequence, triggering the allergic reaction, thereby releasing several pro-inflammatory mediators, such as histamine [5].

Non-IgE-mediated CMA includes different mechanisms than classical IgE-mediated CMA, as they mostly manifest delayed, usually several hours after eating the culprit food, with negative skin prick tests (SPT) and the absence of serum-specific IgE against the suspected allergen. Gastrointestinal symptoms are common, such as diarrhea or constipation, discomfort, reflux, vomiting, and faltering growth. The mechanism is thought to be cell-mediated, involving T helper (Th) cells and other activated lymphocyte populations, such as cytotoxic CD8 T cells. The prevalence of non-IgE-mediated allergies has been described in a study on food hypersensitivity in infants in the UK, and it was demonstrated that the cumulative incidence of non-IgE-mediated allergy to CM was 1.7% [6]. Non-IgE-mediated gastrointestinal food allergic disorders include food protein-induced enterocolitis syndrome (FPIES), food protein-induced allergic proctocolitis (FPIAP), and food protein-induced enteropathy (FPE). An example of mixed IgE-mediated and non-IgE-mediated allergy is eosinophilic esophagitis (EoE) [7].

In many parts of the world, goat milk (GM) has been part of the human diet for millennia. Compared to CM, GM showed greater similarity to human milk, with a softer curd formation in the stomach, better protein digestibility, a higher proportion of small milk fat globules, and different allergenic properties [8]. However, GM contains lower levels of folic acid and vitamin B12 compared to CM. Additionally, GM has higher levels of calcium, phosphorus, and magnesium. As CM lacks the proper amounts of iron, vitamin C, and other nutrients, it is extremely important to only use infant formulas for infants <1 year of age that have been adjusted for these nutrients [9]. GM has long been recommended as an alternative for patients with CMA, but GM proteins have immunological cross-reactivity with CM proteins and should therefore not be used in infants with IgE-mediated allergies. However, in studies with infants suffering from gastrointestinal allergy (non-IgE-mediated) and chronic enteropathy against CM proteins, 40–100% were able to tolerate GM [10]. Currently, there is a growing request for alternative options for infant feeding with potential lower allergenicity. This review will discuss single GM allergy, the allergenicity of GM proteins and their cross-reactivity with CM proteins, GM in IgE- and non-IgE-mediated CMA, and GM as potential interventions for the prevention of milk allergy.

## 2. Single Goat Milk Allergy (Not Associated with CMA)

Single GM allergy, not associated with CMA, is a rare disorder, and there is not much known about the allergenicity of goat milk by itself. Some case reports of a single GM allergy have been described in the literature. The first case was reported in 1999 [11]. It concerned a 2-year-old girl who developed allergic reactions after consuming goat cheese, but not after consuming CM products. Serum IgE was less reactive to CM and its caseins than to goat milk and its caseins. A 4-year-old boy showed gastrointestinal and respiratory symptoms within minutes after consuming a goat/sheep’s milk-derived food product. A SPT was performed 1 month after the allergic reaction and showed it to be particularly positive for GM but negative for CM [12]. Another young child (7-year-old boy) developed anaphylaxis after drinking GM formula [13]. A 27-year-old female patient who could tolerate CM, dairy products, and sheep cheese could not tolerate goat cheese. SPT was positive for goat milk products but negative for CM, and immunoblotting showed an IgE-binding band at 14 kDa. The allergenic protein was thought to be α-lactalbumin (α-LA), which can remain in small amounts during cheese production [14]. A 39-year-old man had repeated episodes of anaphylaxis after eating goat cheese. GM κ-casein was suspected to be the main allergen since the patient had clinical, SPT, and IgE reactions to GM cheese suspected to be caused by κ-casein. It is known that κ-casein has lower interspecies cross-reactivity than other casein proteins, which could explain his tolerance to CM [15].

A retrospective study was performed on 28 children who showed severe allergic reactions after consumption of GM products but could tolerate CM products to characterize the major GM proteins involved in GM allergy. Clinical observations, SPT, and IgE-binding established the diagnosis of a single GM allergy without associated CMA. Enzyme allergosorbent tests showed that GM allergies involve the casein fraction and not whey proteins. CM caseins were not or poorly recognized by the IgE of the patient, while alpha-S1-casein (αS1-casein), alpha-S2-casein (αS2-casein), and beta-casein (β-casein) from GM were recognized with high specificity and affinity. Increasing concentrations of CM caseins showed complete inhibition of GM casein by the IgE of patients and did not inhibit the binding to both sheep and GM caseins. Therefore, it can be concluded that sensitization and clinical allergy to GM, not associated with CMA, are caused by the casein proteins of GM, particularly αS1-, αS2-, and β-caseins [16].

## 3. Goat Milk Protein Allergenicity and Cross-Reactivity with Cow’s Milk Proteins

GM was thought to be a good substitute for CMA individuals for a long time. However, recently it has become clear that goat milk proteins have large homologies with CM proteins (Table 1) [17,18,19] and therefore show cross-reactivity [20,21,22,23,24]. However, there are indications that GM is less allergenic than CM due to the difference in protein composition or species specificity (Figure 1). In a mouse atopy model, CM and GM allergenicities were compared. The number of mice with diarrhea was significantly higher in the CM-sensitized group compared to the GM-sensitized group, with concomitantly significantly higher CM-specific IgG1 and histamine levels in the serum of CM-sensitized mice. It was concluded that when GM was used as a first protein source after breastfeeding, it was less allergenic when compared to CM in mice [25]. Furthermore, a comparison of CM and GM allergenicity was studied in a guinea pig anaphylaxis model. Here, the anaphylaxis and antibody production results also showed that GM is less allergenic when compared to CM [23].

### 3.1. αS1-Casein

Milk caseins, and specifically αS1-casein, are major milk allergens responsible for CMA and seem to play a role in persistent allergies. CM and GM both contain the four main casein classes, i.e., αS1-casein, αs2-casein, β-casein, and κ-casein; however, the composition of these milk proteins is different (Figure 1) [26]. The level of αS1-casein in GM can vary from high (7 g/L), medium (3.2 g/L), low (1.2 g/L), or even absent, which is dependent on the polymorphism of the gene and goat breed, whereas CM always has higher levels (~12 g/L) of αS1-casein. The sequence similarity of αS1-casein in GM is 89.7% when compared with CM [27]. Studies indicate that low levels of αS1-casein in GM may reduce the allergenicity; however, a dose response to αS1-casein has not been established so far. Several in vivo models were conducted to study the sensitizing capacity of αS1-casein. In a mouse model of gastrointestinal allergy, the immune response to different levels of goat αS1-casein was studied. It was found that mice sensitized with αS1-casein had higher levels of IgG1 and IgE antibodies to αS1-casein when compared to the control group. However, groups challenged with different doses of αS1-casein showed similar results. The group challenged with the highest dose of αS1-casein showed 10 times higher levels of mouse mast cell protease-I (MMCP-I), a marker for mast cell degranulation. In cultured splenocytes in the presence of αS1-casein, both interleukin (IL)-4 and IL-10 were produced in a concentration-dependent manner. Thus, milk with lower levels of αS1-casein could contribute to a lesser antigenic burden [27]. Another mouse model studied the sensitizing capacity of αS1-casein from GM compared to CM. The results indicate that CM αS1-casein showed higher allergenicity than GM αS1-casein, as seen by significantly increased αS1-casein-IgE and Th2 cell-related inflammatory mediators. Furthermore, sensitization of CM αS1-casein showed damage to the intestinal barrier in mice and activated the TLR4-NFκB pathway, resulting in an increase in interferon (IFN)-γ [28]. In a mouse model of respiratory allergy, the differences in allergenicity of CM and GM were compared. The GM-sensitized group had a significantly lower number of mice with severe respiratory symptoms when compared to the CM-sensitized group. Serum CM-specific IgE, IgG1, and plasma histamine levels were also lower in GM-sensitized mice. In addition, GM-sensitized mice had lower IL-4 and IL-17A levels and higher IFN-γ and IL-10 levels, indicating a more favorable Th1/Th2 and Treg/Th17 balance [21]. Only one study using human subjects with CMA was conducted. A fresh food SPT with nine GM samples with a low content of αS1-casein was given to six CMA children, which showed no or only weak reactivity to GM samples, whereas all subjects reacted strongly to CM αS1-casein [29].

### 3.2. β-Casein

The major protein in GM is β-casein, which can constitute up to 60% of the casein fraction in GM. Despite a sequence similarity of 91% between GM and CM β-casein, IgE antibodies from GM-allergic but CM-tolerant patients recognize GM β-casein without cross-reacting with CM, which was investigated in a study with 11 CM-allergic subjects and 11 GM-allergic/CM-tolerant subjects. IgE-binding epitopes are spread all over goat β-casein; however, a non-cross-linking version with only five amino acid substitutions of goat β-casein was created and could generate new insights for hypoallergenic modifications [30]. In another study using GM-allergic subjects but tolerant to CM, it was demonstrated that IgE recognized goat β-casein without cross-reaction to cow β-casein. This IgE was mainly directed against the domain 49–79, which differs from bovine β-casein by only three amino acid substitutions [31].

## 4. Goat Milk in IgE-Mediated CMA

For a while, GM has been used as an alternative to CM in CMA patients; however, already in 1939, the first publication on the immunogenic relationship between cow’s and goat’s milk was published, noting that some CMA patients can tolerate GM [10,32], but others cannot [20,24,33,34]. It was concluded that GM definitely has a valuable place in the treatment of some, but certainly not in all infants with CM hypersensitivity [35]. Since then, several reports of anaphylaxis after GM ingestion in CMA patients have been published [24,33,34]. Cross-reactivity was studied in a guinea pig model of CMA, comparing sensitization of CM proteins with either GM proteins with high αS1-casein levels (GM1) or low αS1-casein levels (GM2). Guinea pigs fed CM and GM1 produced high levels of α-β-lg IgG1 (a class of anaphylactic antibodies in guinea pigs), with a distinct cross-reactivity between goat and cow β-LG. However, in GM2-fed guinea pigs, α-β-lg IgG1 antibodies and intestinal anaphylaxis were significantly decreased when compared to GM1-fed guinea pigs. These results suggest that the discrepancies observed in the use of GM in CMA could, at least partly, be attributed to the high genetic polymorphism of GM proteins [36]. The difference in protein fraction between GM and CM was further investigated for cytokine secretion by cultured peripheral blood mononuclear cells (PBMCs) from infants with CMA. After exposure to GM casein and β-LG, the production of the pro-inflammatory tumor necrosis factor (TNF)-α cytokine was lower than after exposure to similar fractions from CM. GM induced greater production of the regulatory cytokine IL-10 by PBMCs than CM. These results show that it is important to first assess the immune reactivity against each protein fraction before considering GM as a safe alternative for infants with CMA [37]. Different breeds of goats (e.g., Jamnapari, Saanen, and Toggenburg) have genetic variances that could influence their milk protein composition and, thereby possibly, their sensitizing capacity. This was investigated in a comparative mapping and identification of allergenic proteins in milk from different dairy goat breeds that cross-reacted with CM allergens. Immunoblotting with IgE from CMA subjects showed that Jamnapari’s milk proteins cross-reacted with four major milk allergens, i.e., αS1-casein, β-casein, κ-casein, and β-LG. On the other hand, milk proteins from Saanen goats cross-reacted with two major milk allergens, i.e., αS1-casein and β-LG, whereas Toggenburg GM proteins cross-reacted with κ-casein only. These findings may provide new insights into the development of hypoallergenic GM by lowering the allergenic milk protein content through cross-breeding of different goat breeds [38].

## 5. Goat Milk in Non-IgE-Mediated CMA

Currently, the management of non-IgE-mediated food allergies involves the avoidance of the culprit food [7], and for 50–70% of cases, it is possible to introduce and use products with baked proteins. There are some indications that GM might be well tolerated in certain patients with non-IgE-mediated CMA, as previous research has shown that 40–100% of these patients can tolerate GM [10]. This seems to be especially true in infants suffering from non-IgE-mediated gastrointestinal allergy and chronic enteropathy since there is no risk for cross-reactive IgE antibody responses [10]. Various anecdotal publications have shown that GM has been used as a hypoallergenic infant food or milk alternative in infants with CMA. A study in 60 infants with marked stool eosinophilia and diarrhea, colic, and/or vomiting showed improvement in symptoms in several cases after consuming GM [39]. Chronic enteropathy in infants caused by CM was resolved by shifting to GM [40]. In another study, around 40% of CMA subjects were able to tolerate GM proteins [41]. Van der Horst suggested that the casein constituents and other protein differences between GM and CM are likely the reason that it can be successfully used in some cases of CMA [42]. However, over the past decades, very little research has been conducted regarding the use of GM in non-IgE-mediated allergy, and controlled clinical studies are needed to identify the patient group(s) that could benefit from GM.

## 6. Goat Milk for Milk Allergy Prevention

Until 2008, guidelines for the prevention of food allergies recommended a delayed introduction of allergenic foods until 3 years of age based on the theory that a lack of exposure to allergenic foods during early infancy would prevent the development of allergies. However, recently, it has become evident that these guidelines may have contributed to the development of food allergies in children. Current guidelines have therefore changed from avoidance to early introduction of allergenic foods already during pregnancy and breastfeeding [43] and early introduction to infants at 4–6 months of age [44].

Due to cross-reactivity, GM proteins can induce allergic reactions in CMA infants, which rules out the recommendation of GM-based formulas for CMA infants. Nevertheless, from animal studies, there is some evidence that GM is less allergenic than CM [21,22,23,25,45], hence the idea of using GM for the prevention of allergic sensitization to milk proteins. The post hoc analysis of a multicentre, double-blind, controlled study for comparison of growth and nutritional status in infants fed GM- and CM-based infant formula allowed to analyze secondary parameter allergy-related outcomes. Atopic dermatitis (AD) was assessed using SCORAD and showed an incidence of 23% in the CM formula group compared to 14% in the GM formula group, although not significant since this study was powered to evaluate growth differences, still suggesting a preventive effect of GM for the onset of AD [46]. Currently, a large study aims to assess a potential reduction in the development of AD by intervention with a GM-based formula compared to a CM-based formula. This two-arm, parallel, randomized, double-blind, controlled intervention study will enroll up to 2296 healthy term-born infants until 3 months. The primary parameter will be the cumulative incidence of AD until 12 months of age, and children will be followed until 5 years of age [47].

The potential role of GM in reducing AD might relate to a beneficial effect on oral tolerance, i.e., the ability of the immune system to tolerate harmless food proteins, as opposed to the elimination of pathogens. Oral tolerance depends on allergen exposure in an appropriate immune environment [48]. For example, for CM, it has been investigated that intact whey proteins can induce allergic sensitization, whereas specific epitopes (tolerogenic peptides) in hydrolyzed whey proteins can support the development of oral tolerance to whey [49,50]. One can hypothesize, since GM proteins are associated with faster [51] and more efficient [52] digestibility than CM proteins, that these tolerogenic peptides might naturally occur during early digestion, thereby avoiding the presence of allergenic epitopes and favoring the production of tolerogenic peptides. However, tolerance is in a certain way the opposite of sensitization, a phenomenon probably largely related to alterations of the epithelial barrier at multiple sites, globally favored by genetic predisposition and environmental factors [53]. Recent studies indicate that the gut microbiome composition influences the development of the immune system and modulates immune mediators, which successively can influence the intestinal barrier [54]. Goat milk can enhance innate and adaptive immune responses and reduce allergen-induced airway inflammation in mice offspring [55]. A recent study assessed the shaping of the gut microbiota and the formed metabolites by GM-based infant formula in mice colonized by the feces of healthy infants [56]. A potential beneficial effect of GM on the development of allergies will need to take into consideration all these contributing factors.

## 7. Conclusions

Allergy to GM, not associated with CMA, is a rare disorder, and only a few cases of a single GM allergy have been reported. GM proteins have immunological cross-reactivity with CM proteins due to the large homology between the milk proteins and should therefore not be used in infants with IgE-mediated CMA. There are indications that GM could be of benefit in certain types of non-IgE-mediated allergy or even in the prevention of sensitization to milk proteins, but controlled clinical studies are needed to confirm this hypothesis. However, due to its lower allergenicity, GM might be a better choice over CM as a first source of protein when breastfeeding is not possible or after the breastfeeding period. The preventive effect of GM on allergy symptoms, such as AD or CMA, will need further research.

## Figures and Tables

**Figure 1 nutrients-16-02402-f001:**
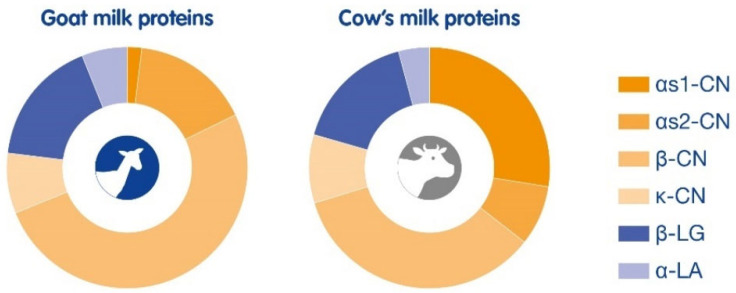
Composition of goat milk proteins compared to cow’s milk proteins. Goat milk protein distribution: αS1-CN 1%, αS2-CN 16%, β-CN 51%, κ-CN 8%, β-LG 17%, and α-LA 6%. Cow’s milk protein distribution: αS1-CN 27%, αS2-CN 8%, β-CN 34%, κ-CN 9%, β-LG 16%, and α-LA 4%.

**Table 1 nutrients-16-02402-t001:** Homology of goat milk proteins compared to cow’s milk proteins [17,18,19].

Milk Protein	Homology of Goat Milk Proteins Compared to Cow’s Milk Proteins
αS1-casein	89.7%
αS2-casein	90.1%
β-casein	91.0%
κ-casein	59.5%
β-lactoglobulin	88.0%
α-lactalbumin	92.0%

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
