# Peer review of "Goat Milk Allergy and a Potential Role for Goat Milk in Cow’s Milk Allergy"

_nutrients, 2024, doi:10.3390/nu16152402_

Round 1

Reviewer 1 Report

Comments and Suggestions for Authors

An interesting and important manuscript from a practical point of view. A proper introduction, a thorough background of the literature.
I believe that in order to increase the quality of work, it would be good to analyze the occurrence of allergies depending on the usual consumption of cow's milk. Are these values ​​similar or completely different in Scandinavian countries and Japan?
Figure 1 - please correct the figure - it is not 600 dpi!, and the legend font is a few points larger
I believe that the work is not only interesting but worth publishing quickly

Author Response

An interesting and important manuscript from a practical point of view. A proper introduction, a thorough background of the literature. I believe that the work is not only interesting but worth publishing quickly. The authors would like to thank the reviewer for expressing the interest of our manuscript

Comment 1: I believe that in order to increase the quality of work, it would be good to analyze the occurrence of allergies depending on the usual consumption of cow's milk. Are these values ​​similar or completely different in Scandinavian countries and Japan? This is a very interesting point of view as the prevalence CMA could be influenced by how commonly cow’s milk is consumed in certain parts of the world. However, a review I found show that there are differences in certain food allergies such as peanut and wheat, but the prevalence of CMA is comparable worldwide. This is added to the introduction, 2nd sentence ‘Although there are differences in prevalences of food allergies such as peanut and wheat in different parts of the world, prevalence CMA is comparable worldwide [2].’

Figure 1 - please correct the figure - it is not 600 dpi!, and the legend font is a few points larger. A complete new figure has been designed and the figure legend is adjusted.

Reviewer 2 Report

Comments and Suggestions for Authors

The authors conducted an interesting and detailed review on the role of goat milk in non-IgE mediated allergy and the prevention/induction of oral tolerance in milk allergy. While the protein composition of goat milk is similar yet distinct from cow milk, making it less allergenic, it is important to note some drawbacks that should be analyzed and highlighted in the text. For instance, goat milk contains lower levels of folic acid and vitamin B12 compared to cow milk, necessitating supplementation. Additionally, goat milk has higher levels of calcium, phosphorus, and magnesium, which can determine issues for the kidneys of infants and very young children who may not be able to manage the mineral load. Furthermore, goat milk has a higher content of saturated fats, which might not be ideal for a balanced diet (Stergiadis S, Nørskov NP, Purup S, Givens I, Lee MRF. Comparative Nutrient Profiling of Retail Goat and Cow Milk. Nutrients. 2019 Sep 24;11(10):2282. doi: 10.3390/nu11102282).

The section on prevention should be further enriched by referencing current strategies for food allergy prevention not only during early childhood (Trogen B, Jacobs S, Nowak-Wegrzyn A. Early Introduction of Allergenic Foods and the Prevention of Food Allergy. Nutrients. 2022 Jun 21;14(13):2565. doi: 10.3390/nu14132565), but also during pregnancy (Manti S, Galletta F, Bencivenga CL, Bettini I, Klain A, D'Addio E, Mori F, Licari A, Miraglia Del Giudice M, Indolfi C. Food Allergy Risk: A Comprehensive Review of Maternal Interventions for Food Allergy Prevention. Nutrients. 2024 Apr 8;16(7):1087).

Comments on the Quality of English Language

The authors conducted an interesting and detailed review on the role of goat milk in non-IgE mediated allergy and the prevention/induction of oral tolerance in milk allergy. While the protein composition of goat milk is similar yet distinct from cow milk, making it less allergenic, it is important to note some drawbacks that should be analyzed and highlighted in the text. For instance, goat milk contains lower levels of folic acid and vitamin B12 compared to cow milk, necessitating supplementation. Additionally, goat milk has higher levels of calcium, phosphorus, and magnesium, which can determine issues for the kidneys of infants and very young children who may not be able to manage the mineral load. Furthermore, goat milk has a higher content of saturated fats, which might not be ideal for a balanced diet (Stergiadis S, Nørskov NP, Purup S, Givens I, Lee MRF. Comparative Nutrient Profiling of Retail Goat and Cow Milk. Nutrients. 2019 Sep 24;11(10):2282. doi: 10.3390/nu11102282).

The section on prevention should be further enriched by referencing current strategies for food allergy prevention not only during early childhood (Trogen B, Jacobs S, Nowak-Wegrzyn A. Early Introduction of Allergenic Foods and the Prevention of Food Allergy. Nutrients. 2022 Jun 21;14(13):2565. doi: 10.3390/nu14132565), but also during pregnancy (Manti S, Galletta F, Bencivenga CL, Bettini I, Klain A, D'Addio E, Mori F, Licari A, Miraglia Del Giudice M, Indolfi C. Food Allergy Risk: A Comprehensive Review of Maternal Interventions for Food Allergy Prevention. Nutrients. 2024 Apr 8;16(7):1087).

Author Response

The authors conducted an interesting and detailed review on the role of goat milk in non-IgE mediated allergy and the prevention/induction of oral tolerance in milk allergy. The authors want to thank the reviewer for his extensive review and suggestions to improve the manuscript.

Comment 1: While the protein composition of goat milk is similar yet distinct from cow milk, making it less allergenic, it is important to note some drawbacks that should be analyzed and highlighted in the text. For instance, goat milk contains lower levels of folic acid and vitamin B12 compared to cow milk, necessitating supplementation. Additionally, goat milk has higher levels of calcium, phosphorus, and magnesium, which can determine issues for the kidneys of infants and very young children who may not be able to manage the mineral load. Furthermore, goat milk has a higher content of saturated fats, which might not be ideal for a balanced diet (Stergiadis S, Nørskov NP, Purup S, Givens I, Lee MRF. Comparative Nutrient Profiling of Retail Goat and Cow Milk. Nutrients. 2019 Sep 24;11(10):2282. doi: 10.3390/nu11102282). Added to the introduction 4th paragraph:  However, GM contains lower levels of folic acid and vitamin B12 compared to CM. Additionally, GM has higher levels of calcium, phosphorus, and magnesium. As CM lacks the proper amounts of iron, vitamin C and other nutrients, it is extremely important to only use infant formulas in infants <1 year of age that have been adjusted for these nutrients [8].

Comment 2: The section on prevention should be further enriched by referencing current strategies for food allergy prevention not only during early childhood (Trogen B, Jacobs S, Nowak-Wegrzyn A. Early Introduction of Allergenic Foods and the Prevention of Food Allergy. Nutrients. 2022 Jun 21;14(13):2565. doi: 10.3390/nu14132565), but also during pregnancy (Manti S, Galletta F, Bencivenga CL, Bettini I, Klain A, D'Addio E, Mori F, Licari A, Miraglia Del Giudice M, Indolfi C. Food Allergy Risk: A Comprehensive Review of Maternal Interventions for Food Allergy Prevention. Nutrients. 2024 Apr 8;16(7):1087). The authors agree that the part of current strategies for allergy prevention could be more elaborated, the refs are added and in the heading Goat milk for milk allergy prevention as a first paragraph is added: Until 2008, guidelines for prevention of food allergy recommended a delayed introduction of allergenic foods until 3 years of age based on the theory that lack of exposure to allergenic foods during early infancy would prevent the development of allergy. However, recently it has become evident that these guidelines may have contributed to the development of food allergies in children. Current guidelines have therefore changes from avoidance to early introduction of allergenic foods already during pregnancy and breastfeeding period [43], and early introduction to infants 4-6 months of age [44].

Reviewer 3 Report

Comments and Suggestions for Authors

Dear Sirs,

This review article “Goat milk allergy and a potential role of goat milk in non-IgE mediated milk allergy or milk allergy prevention/oral tolerance induction” by Benjamin-van Aalst et al,  is very well written, making it very pleasant to read and easy to comprehend. It covers a wide range of the “milk allergy” topic, which is very “popular” among pediatricians and parents and gives important information without being too hard to follow, even for someone not so involved in the specific topic. The abstract and the conclusions, even though quite similar, are very concise.

If some exist, the authors could add information about studies on goat milk and a history of cow milk allergy in the family. It is a very “hot” topic, taking into account that “milk allergy” is probably overdiagnosed/overtreated in everyday pediatrics.

1.     The title is not very clear (confusing and doesn’t include IgE-mediated allergy).

2.     Line 56. It needs rephrasing, so the meaning is clear that the cumulative incidence of non-IgE mediated allergy to CM was 1,7%

3.     Line 79: than to goat milk and “its” caseins

4.     Line 104: after the reference [14] at the end of the paragraph, figure 1 is depicted in parentheses (which itself has reference 24 in its title). Please make sure this is correct, and if yes, it probably needs rephrasing to be clearer.

5.     Figure 1: the letters in the figure are not clear

6.     Figure 1: the reference doesn’t contain the specific figure, please explain better the way it was constructed

7.     Line 199-205: Maybe a sentence about the different breeds of goats would be helpful, before only stating the different names (ie Toggenburg, Saanen, etc).

8.     Line 211: “This seems to be especially true in infants…”

9.     Line 232-234: Atopic dermatitis incidence of 24% seems extremely high and could not be found in reference 42. Please check again.

10.  References: need correction as to their format

Comments on the Quality of English Language

Minor editing of English language required

Author Response

This review article “Goat milk allergy and a potential role of goat milk in non-IgE mediated milk allergy or milk allergy prevention/oral tolerance induction” by Benjamin-van Aalst et al,  is very well written, making it very pleasant to read and easy to comprehend. It covers a wide range of the “milk allergy” topic, which is very “popular” among pediatricians and parents and gives important information without being too hard to follow, even for someone not so involved in the specific topic. The abstract and the conclusions, even though quite similar, are very concise. The authors would like to thank the reviewer for his extensive review and suggestions to improve the paper.

If some exist, the authors could add information about studies on goat milk and a history of cow milk allergy in the family. It is a very “hot” topic, taking into account that “milk allergy” is probably overdiagnosed/overtreated in everyday pediatrics. I have approached many expert in the field of CMA and found that many have tried goat milk (IF) at certain times, but have never been studied in a good defined clinical setting and the evidence is therefore what we call ‘anecdotal’. However, I’m happy to hear such studies are of interest as I am in discussion with some professionals how to conduct these kind of studies that could be incorporated in current guidelines. Therefore I hope to be able to provide data on studies with goat milk in CMA soon.

Comment 1.     The title is not very clear (confusing and doesn’t include IgE-mediated allergy). The title is changed to ‘Goat milk allergy and its potential role in cow’s milk allergy’

Comment 2.     Line 56. It needs rephrasing, so the meaning is clear that the cumulative incidence of non-IgE mediated allergy to CM was 1,7%. Rephased to: Prevalence of non-IgE mediated allergies has been described in a study on food hyper-sensitivity in infants in the UK and it was demonstrated that the cumulative incidence of non-IgE-mediated allergy to CM was 1.7%.

Comment 3.     Line 79: than to goat milk and “its” caseins. Adjusted

Comment 4.     Line 104: after the reference [14] at the end of the paragraph, figure 1 is depicted in parentheses (which itself has reference 24 in its title). Please make sure this is correct, and if yes, it probably needs rephrasing to be clearer. The phrase ‘Figure 1’ was incorrect here and is deleted

Comment 5.     Figure 1: the letters in the figure are not clear. A new figure was designed and added

Comment 6.     Figure 1: the reference doesn’t contain the specific figure, please explain better the way it was constructed. The figure legend is adjusted with added distribution % from the ref.

Comment 7.     Line 199-205: Maybe a sentence about the different breeds of goats would be helpful, before only stating the different names (ie Toggenburg, Saanen, etc). Rephrased to: Different breeds of goats (e.g. Jamnapari, Saanen, Toggenburg) have genetic variances that could affect their milk composition and thereby possibly also their sensitizing capacity. This was shown is a comparative mapping and identification of allergenic proteins in milk from different dairy goat breeds that cross-reacted with cow’s milk allergens. Analysis of IgE-reactive proteins of CMA patients revealed that Jamnapari's milk proteins were found to cross-react with four major milk allergens: αS1-casein, β-casein, κ-casein, and β-LG. On the other hand, Saanen goat's milk proteins cross-reacted with two major milk allergens, αS1-casein and β-LG, whereas Toggenburg GM proteins only react with κ-casein.

Comment 8.     Line 211: “This seems to be especially true in infants…” ‘true’ is added

Comment 9.     Line 232-234: Atopic dermatitis incidence of 24% seems extremely high and could not be found in reference 42. Please check again. The authors agree that 23% prevalence of AD is high and he references [42 + 43] and were checked again, however similar statement was found in [43] but we’ve added ‘not significant’ to the sentence: The post hoc analysis of a multicentre, double-blind, controlled feeding study for com-parison of growth and nutritional status in infants fed GM- and CM-based infant formula allowed to analyse secondary parameter allergy-related outcomes. Atopic dermatitis (AD) assessed using SCORAD had an incidence of 23% in the CM formula group compared with 14% in the GM formula group, although not significant since the study was powered to evaluate growth differences, still suggesting a preventive effect of GM for the onset of AD [42].

Comment 10: References: need correction as to their format. The reference list needs to be updated since reviewers requested addition of references, will correct the format in collaboration with the journal